# An Edge Intelligent Method for Bearing Fault Diagnosis Based on a Parameter Transplantation Convolutional Neural Network

**Xiang Ding** [1], **Hang Wang** [1,2], **Zheng Cao** [1], **Xianzeng Liu** [1,2], **Yongbin Liu** [1,2,*] and **Zhifu Huang** [1]

1   School of Electrical Engineering and Automation, Anhui University, Hefei 230601, China;
    z20301128@stu.ahu.edu.cn (X.D.); hangwang@ahu.edu.cn (H.W.); caozheng@ahu.edu.cn (Z.C.);
    liuxianzeng@ahu.edu.cn (X.L.); z20301126@stu.ahu.edu.cn (Z.H.)
2   Anhui Joint Key Laboratory of Energy Internet Digital Collaborative Technology, Hefei 230088, China
*   Correspondence: lyb@ustc.edu.cn

**Abstract:** A bearing is a key component in rotating machinery. The prompt monitoring of a bearings' condition is critical for the reduction of mechanical accidents. With the rapid development of artificial intelligence technology in recent years, machine learning-based intelligent fault diagnosis (IFD) methods have achieved remarkable success in the field of bearing condition monitoring. However, most algorithms are developed based on computer platforms that focus on analyzing offline, rather than real-time, signals. In this paper, an edge intelligence diagnosis method called S-AlexNet, which is based on a parameter transplantation convolutional neural network (CNN), is proposed. The method deploys the lightweight IFD method in a low-cost embedded system to monitor the bearing status in real time. Firstly, a lightweight IFD algorithm model is designed for embedded systems. The model is trained on a PC to obtain optimal parameters, such as the model's weights and bias. Finally, the optimal parameters are transplanted into the embedded system model to identify the bearing status on the edge side. Two datasets were used to validate the performance of the proposed method. The validation using the CWRU dataset shows that the proposed method achieves an average prediction accuracy of 94.4% on the test set. The validation using self-built data shows that the proposed method can identify bearing operating status in embedded systems with an average prediction accuracy of 99.81%. The results indicate that the proposed method has the advantages of high recognition accuracy, low model complexity, low cost, and high portability, which allow for the simple and effective implementation of the edge IFD of bearings in embedded systems.

**Keywords:** edge computing; intelligent fault diagnosis; CNN; bearings; embedded systems





## 1. Introduction

Rolling bearings are key components in rotating machinery. According to statistics, more than 30% of rotating machinery failures are the result of bearing failure [1]. Therefore, detecting the bearing operating condition in real time is crucial for the reduction of potential mechanical accidents and economic losses brought about by bearing failure, while ensuring the safe running of machinery.

Existing fault diagnosis methods for rolling bearings can be broadly divided into two categories: model-driven and data-driven. Prior to the 1980s, rolling bearing fault diagnosis was implemented using knowledge models [2]. The common processing method involves the determination of the bearing fault type by combining multiple statistical indicators or the conversion of the time domain signal of the bearing into a frequency domain signal in order to determine the spectral value, such as with the fast Fourier transform, wavelet transform, etc. However, the Fourier transform is a whole transformation, which lacks the time domain localization, and the wavelet transform requires large computations, making it difficult to realize real-time processing.

With the rapid growth of computing power in recent years, the development of data-driven fault diagnosis methods has also been promoted [3,4]. Data-driven methods

construct a nonlinear mapping from the fault dimension to the feature dimension by learning large amounts of bearing data with known faults, without relying on additional prior knowledge and expert experience. In most cases, the verification of some machine learning methods, such as random forest [5], support vector machines (SVM) [6], and principal component analysis (PCA) [7], is facilitated by the possibility to feed enough machine status data to the learning system in order for it to learn the characteristics of the data. However, these shallow neural networks are not effective in learning complex nonlinear mapping relations due to their limited learning capabilities. Deep learning (DL), which automatically learns the hierarchical features and correlations between data [8,9], has been widely used in various fields. Currently, DL is widely used in fault diagnosis due to its powerful feature learning capability [10,11]. Numerous deep learning methods, such as generative adversarial network (GAN) [12], convolutional neural network (CNN) [13,14], and recurrent neural network (RNN) [15], have been applied in fault diagnosis, with CNNs being the most widely used. Wang et al. proposed a multiscale CNN with a joint one-dimensional (1D) and two-dimensional (2D) feature extraction function that can distinguish correlations between adjacent and non-adjacent intervals in periodic signals [16]. Xu et al. used CNN to extract features from the components of the variable mode decomposition (VMD) method for bearing fault diagnosis [17]. Yang et al. performed bearing fault diagnosis by preprocessing the original signal with three different data processing methods and fusing the outputs of four CNNs with the use of a fuzzy fusion strategy [18]. Cheng et al. proposed a method that used local binary convolutional layers instead of traditional convolutional layers and combined it with the continuous wavelet transform (CWT), which could effectively diagnose both bearing and gearbox compound faults [19]. Fang et al. proposed a lightweight model for rotating machinery diagnosis based on dynamic convolution and separable convolution strategies [20]. Ji et al. proposed an order-tracking method using 1DCNN in two steps for the fault diagnosis of variable condition signals [21]. Gao et al. proposed the use of hierarchically trained CNNs to overcome the problem of unbalanced data distribution during fault diagnosis [22]. Therefore, it is evident that PC-based CNNs have been widely applied in fault diagnosis. For this paper, the classical CNN network AlexNet has been selected to extract the bearing features and identify the bearing states.

With the rapid development of the internet and the internet of things, mechanical devices in industrial sites are continuously generating high-speed real-time data at an unprecedented rate, such as temperature, humidity, audio, video, etc. [23,24]. Data volume is gradually increasing, data types are more diverse, and data structures are increasingly more complex. Meanwhile, fault diagnosis for rotating machinery involves the following limitations. First, the scale of the data is too huge to upload to the server and be analyzed. Second, the data often have low information density. As the mechanical equipment usually works in normal conditions, the data generated have a great deal of redundancy, which makes it necessary to extract useful information from the data. Third, the timeliness of the data is extremely important. Whether it is a small machine for industrial production or an aircraft engine, both are tightly constructed machines, and the failure of a component (such as bearings) will quickly result in huge economic losses. Therefore, it is essential to analyze and process the data as soon as possible in order to provide an efficient diagnosis and a timely warning. Edge computing can analyze the real-time data directly on the edge side, thus avoiding data congestion and a delay in diagnosis caused by the uploading of data to the server [25,26]. For bearing fault diagnosis, IFD models deployed in embedded systems to identify the bearing status on the edge side not only reduce the network bandwidth pressure and energy consumption caused by the uploading of data to the server, but they also guarantee the timeliness of the data. They can help detect the abnormal status of a bearing in real time and prevent the loss of life and property caused by the fault.

So far, IFD technologies, based on signal analysis and machine learning, have achieved remarkable success in the field of condition monitoring of mechanical equipment [27–29]. However, most of the research has been conducted on a PC, and it is difficult to deploy

and execute IFD models in embedded systems due to their computational power, running memory, and storage capacity [30]. Lu et al. implemented the inference and result display of the fault diagnosis algorithm using a stochastic resonance-based adaptive filter with two STM32F4 series MCUs [31]. Lu et al. used two microcontroller units to synchronously collect the phase information and vibration signals of motor bearings in order to implement online fault diagnosis based on order analysis in embedded systems [32]. Pham et al. built a CNN-based MobileNet-v2 model and transplanted it to Raspberry Pi 3 to carry out the fault classification of bearings [33]. Chen et al. proposed a lightweight fault diagnosis system that employed the random forest algorithm and Hilbert transform on the Xilinx PYNQ-Z2 development board [34]. Park et al. proposed LiReD, a lightweight, single-board, computer-based, real-time fault detection system, which consisted of two parts: a front-end for real-time monitoring based on Raspberry Pi 3 and a back-end for the training of LSTM-based networks [35]. Most current approaches to deploying IFD models in embedded systems are based on high-performance microcontrollers, such as Raspberry Pi, which usually support machine learning languages, such as Python, thus facilitating the deployment of IFD models in microcontrollers, However, the convenience also comes with increased cost and limited applicability. Thus, deploying the IFD model on a more general and low-cost platform is a way to make edge IFD more widely applicable.

An edge intelligence diagnosis method for bearing faults, called S-AlexNet, which is based on a parameter transplantation CNN, is proposed in this paper [36]. This work aims to deploy IFD models in low-cost MCUs in order to implement the monitoring and fault diagnosis of bearing status on the edge side. This method allows for onsite and real-time monitoring and identification of the bearing status by analyzing the obtained bearing vibration signals on the edge. Unlike traditional PC-based CNN [37] fault diagnosis methods, this paper uses intelligent fault diagnosis algorithms deployed in embedded systems to achieve real-time bearing monitoring [38]. By being close to the edge, the embedded system can greatly reduce the transfer pressure and cloud computing power consumption for data uploads to the cloud, while improving the real-time diagnostic performance. This method is highly applicable in electromechanical equipment fault diagnosis, particularly where it is inconvenient to transfer data to servers or where a real-time diagnosis is required.

The main technical contributions of this work are summarized as follows.

1. A lightweight CNN model, called S-AlexNet, which is easier to deploy in embedded systems, is proposed in this paper.
2. The S-AlexNet-based IFD model is deployed in embedded systems to identify the operating state of bearings on the edge side.
3. In the proposed method, model deployment does not require hardware to support artificial intelligence languages, such as Python, which greatly reduces the hardware cost and expands the application of the proposed method.

The rest of the paper is organized as follows: Section 2 presents the embedded system-based CNN model. Section 3 outlines the parameter training and transplantation methods for the embedded neural network models. Section 4 validates the effectiveness of the proposed method through experimental data. Finally, our conclusions are summarized in Section 5.

## 2. A CNN Model Based on Embedded Systems

### 2.1. Proposed Model

Embedded systems have some advantages, such as small size, low power consumption, and low cost, which allow for the direct application of algorithms on the edge side. However, due to small cores, insufficient memory, and limited computing power, embedded systems take a lot of time when processing complex models with multiple parameters, which makes the system response slower [39]. Therefore, designing lightweight models is essential for embedded systems [40]. The network structure of AlexNet was deepened based on the classic convolutional neural network LeNet5 [41]. AlexNet improved the

competition among neurons, reduced the occurrence of overfitting, increased the model's ability to generalize, and more accurately identified the local characteristics of the data. Moreover, compared to other deep learning methods, AlexNet has a simpler architecture and requires fewer parameters. In this paper, a simplified AlexNet (S-AlexNet) model is proposed by adjusting the number of network channels and fully connected layers. Taking a fault prediction task with *N* classification as an example, the detailed structural parameters of AlexNet and S-AlexNet are shown in Table 1. As shown in Table 1, S-AlexNet has a more compact structure than AlexNet, with only 11 network layers and a greatly reduced number of parameters, making it more suitable for embedded systems. The S-AlexNet model for intelligent fault diagnosis is shown in Figure 1.

**Table 1.** Comparison of the structure of AlexNet and S-AlexNet.

| Layer | Name | AlexNet | | | S-AlexNet | | |
| | | Layer | Output Form | Parameters | Layer | Output Form | Parameters |
|---|---|---|---|---|---|---|---|
| 1 | | Input | $3 \times 224 \times 224$ | 0 | Input | $1 \times 32 \times 32$ | 0 |
| 2 | | Convolution 1 | $96 \times 55 \times 55$ | 34,944 | Convolution 1 | $4 \times 32 \times 32$ | 404 |
| 3 | | Max pooling 1 | $96 \times 27 \times 27$ | 0 | Max pooling 1 | $4 \times 15 \times 15$ | 0 |
| 4 | | Convolution 2 | $256 \times 27 \times 27$ | 614,656 | Convolution 2 | $4 \times 15 \times 15$ | 404 |
| 5 | | Max pooling 2 | $256 \times 13 \times 13$ | 0 | Max pooling 2 | $4 \times 7 \times 7$ | 0 |
| 6 | | Convolution 3 | $384 \times 13 \times 13$ | 885,120 | Convolution 3 | $8 \times 7 \times 7$ | 296 |
| 7 | | Convolution 4 | $384 \times 13 \times 13$ | 1,327,488 | Convolution 4 | $8 \times 7 \times 7$ | 584 |
| 8 | | Convolution 5 | $256 \times 13 \times 13$ | 884,992 | Convolution 5 | $8 \times 7 \times 7$ | 584 |
| 9 | | Max pooling 3 | $256 \times 6 \times 6$ | 0 | Max pooling 3 | $8 \times 3 \times 3$ | 0 |
| 10 | | Fully connected 1 | $1 \times 1 \times 4096$ | 37,748,736 | Fully connected 1 | $1 \times 1 \times 64$ | 4672 |
| 11 | | Fully connected 2 | $1 \times 1 \times 4096$ | 16,777,216 | Output | $1 \times 1 \times N$ | $64 \times N + N$ |
| 12 | | Output | $1 \times 1 \times N$ | $4096 \times N + N$ | — | — | — |
| Total | | — | — | $5.8 \times 10^7 +$ $4097 \times N$ | — | — | $6944 + 65 \times N$ |

In general, the vibration signal collected by the sensor is one-dimensional, which is not suitable for input into the CNN and usually contains noise. Therefore, data pre-processing of the original signal is required. Considering the excellent performance of CNN in the field of image recognition, the strategy of converting one-dimensional vibration signals into two-dimensional images is adopted. The vibration signal can be transformed into many types of images, such as time–frequency plots, recurrence plots, Gramian angular fields, etc. However, these conversion methods involve too many floating-point operations, which will reduce the response speed of the embedded system and are not suitable for the edge IFD scenario proposed in this paper. This study uses a method that converts one-dimensional (1D) time-domain signals into two-dimensional (2D) grayscale images. The method requires little floating-point operations to convert one-dimensional vibration signals into two-dimensional images and reduces the impact of noise on model recognition accuracy [42].

The specific process is shown in Figure 2: suppose the signal acquired by the sensor for a segment is *X* (length *L*). *X* is truncated into *n* segments of length *m* and written as *X* = {$X_1$, $X_2$, $X_3$, ... , $X_n$}, *n* = int (*L*/*m*). Each truncated signal is converted into a two-dimensional grey image [43,44] in the following order:

$$P_{u,v}^r = \frac{X_r(k \times u + v) - \min(X_r)}{\max(X_r) - \min(X_r)} \tag{1}$$

where $P^r_{u,v}$ represents the size of the pixel value in the row $u$ and column $v$ of the $r$-th grayscale image, $u, v = 1, 2, 3, \ldots, k$, $k$ represents the length and width of the grayscale image. As the sensitivity of the model to the aspect ratio of the image is unknown, the image is taken to be a square as a compromise. So, $k = \sqrt{m}$.

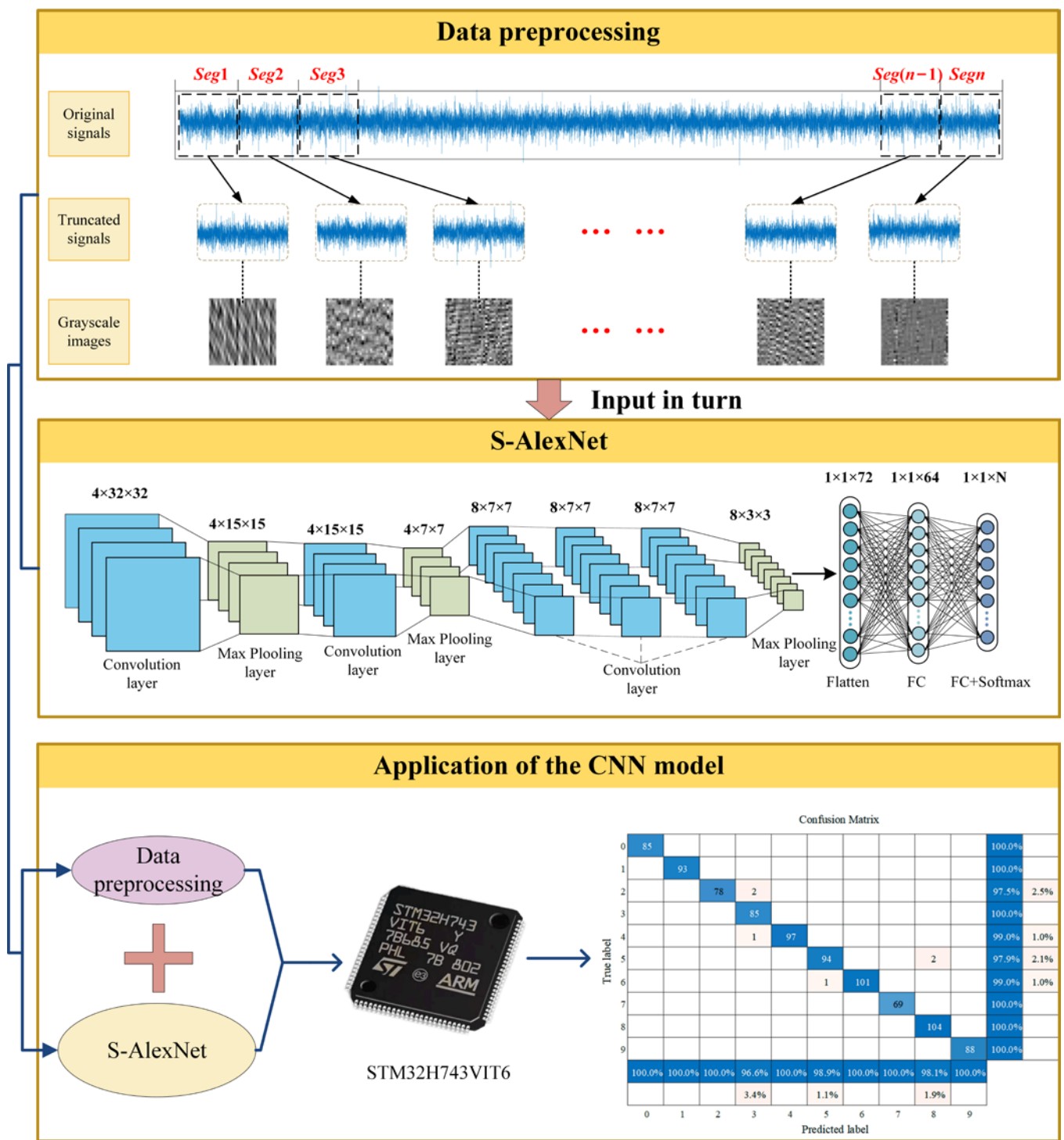

**Figure 1.** Intelligent fault diagnosis CNN model.

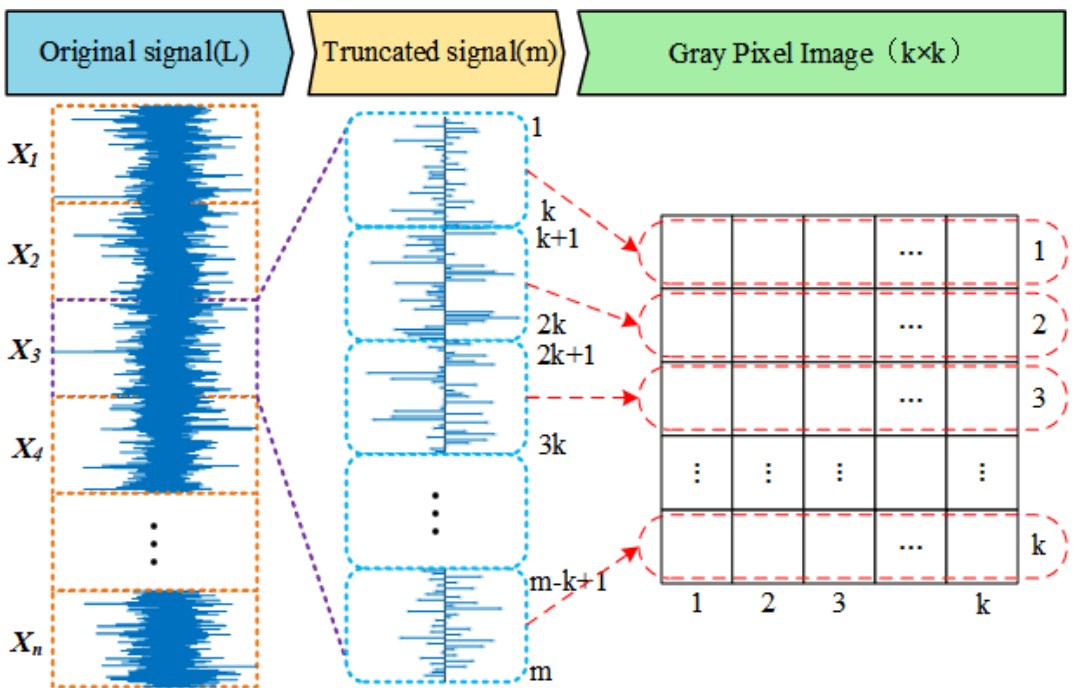

**Figure 2.** Signal-to-image conversion method.

The S-AlexNet model is constructed with a comprehensive consideration of issues, such as resource occupation and computational power, as shown in Figure 1. The model consists of two successive sets of convolution-pooling layers, two convolutional layers, one convolution-pooling layer, and one fully-connected layer.

The grayscale images $P^r$ are input into the network consecutively. Firstly, it is mapped to the input of the next layer by the convolution layer. The output of the convolution layer is expressed as follows:

$$y^r_{i,j} = \sum_{p=1}^{s} \sum_{q=1}^{t} W^z_{p,q} \times P^r_{i-p+1,j-p+1} + b^z_0 \tag{2}$$

where $y^r_{i,j}$ is the value of the row $i$ and column $j$ of the output matrix, $W^z_{p,q} \in R_{s \times t}$ is the weight of the row $p$ and column $q$ of the convolution kernel, $s < k, t < k$, $b^z_0$ is generally the bias of the convolution kernel, and $z$ is the number of channels. The zero-padding method is applied to all the convolutional layers in order to control the feature size. In this method, zeros are added to the input for the corresponding number of rows or columns, according to Equation (3). This equation for calculating the number of rows and columns of complementary zeros can be expressed as follows:

$$\left.\begin{array}{l} P_h = (G_h - 1) \times S_h + W_h - F_h \\ P_w = (G_w - 1) \times S_w + W_w - F_w \end{array}\right\} \tag{3}$$

where $G$ is the output image, $S$ is the step length, $W$ is the convolution kernel, $F$ is the input image, and $h$, $w$ represent the height and width, respectively. The *Relu* function enables nonlinear transformations and is one of the most widely used activation functions in CNNs. The output of the convolutional layer obtained by *Relu* activation can be expressed as follows:

$$g_{i,j}^r = Relu(y_{i,j}^r) = \max\left\{0, y_{i,j}^r\right\} \tag{4}$$

where $g_{i,j}^r$ is the value of the row $i$ and column $j$ of the output matrix. The activation layer output is sampled with the pooling layer to reduce dimensionality and complexity. Subsequently, the max pooling method is selected, i.e., the maximum value of the image region is chosen as the output value after region pooling:

$$Z_{m,n}^r = \max(g_{i,j}^r) \tag{5}$$

where $Z_{m,n}^r$ represents the pooled output of the region $R_{m,n}$, $R_{m,n} \subseteq R_{i,j}$, and $g_{i,j}^r$ represents the previous activation layer output in the region $R_{m,n}$. After two convolution-pooling mappings according to Equations (2), (4), and (5), two successive convolutional layers are used to enhance the features. Then, the output of the previous layer is subjected to a convolution-pooling mapping, according to Equations (2), (4), and (5). The results are then expanded to a feature vector $O^r = \left\{O_1^r, O_2^r, O_3^r, \ldots, O_L^r\right\}$. Furthermore, $O^r$ goes through a fully connected layer and a Sigmoid activation function to obtain the $i$-th output of the $N$ classification model:

$$Y(i) = \frac{1}{1 + e^{-\left(\sum\limits_{i=1}^{N}\sum\limits_{j=1}^{L} W_{i,j}^z \times O_j^r + b_1^z(i)\right)}} \tag{6}$$

where $W_{i,j}^z$ is the weight of the row $i$ and column $j$ of the weight matrix, and $b_1^z(i)$ is the bias of the $i$-th output of the fully connected layer. Finally, the $Y(i)$ output is converted into labels 1~$N$ using a SoftMax layer. In a specific classification issue, $N$ corresponds to the number of known types of bearing states in the target data.

### 2.2. Model Hyper-Parameters

The model uses a large convolutional kernel of $10 \times 10 \times 4$ between the input and the first convolutional layer to obtain a wide perceptual field. The size of the convolution kernel in the second convolution layer is then reduced to $5 \times 5 \times 4$. The convolution kernel is adjusted to $3 \times 3 \times 8$, starting from the third convolution layer, in order to reduce the parameter number and extract more features. The *Relu* function was selected as the activation function for all the convolutional layers in the network in order to increase the non-linear relationship between the layers. At the same time, this produces an output of zero for some neurons, which makes the network sparse, reduces the dependency on parameters, and alleviates the overfitting problem. Max pooling is utilized for all the pooling layers (sized $3 \times 3$ with a step size of 2) to reduce the dimensions and the number of parameters, which simplifies the network.

## 3. Parameter Training and Transplantation

CNN training requires significant powerful computing power and a large storage space. While the current state-of-the-art, high-performance embedded systems can also train the network, this capability does not improve efficiency and significantly increases cost. On the other hand, computers are well suited for artificial intelligence computing due to their powerful GPU computing power. Therefore, the model training in this paper is conducted on a PC, and the optimal parameters after training are transplanted to the network built in the embedded system. The model parameter training and transplantation process are shown in Figure 3.

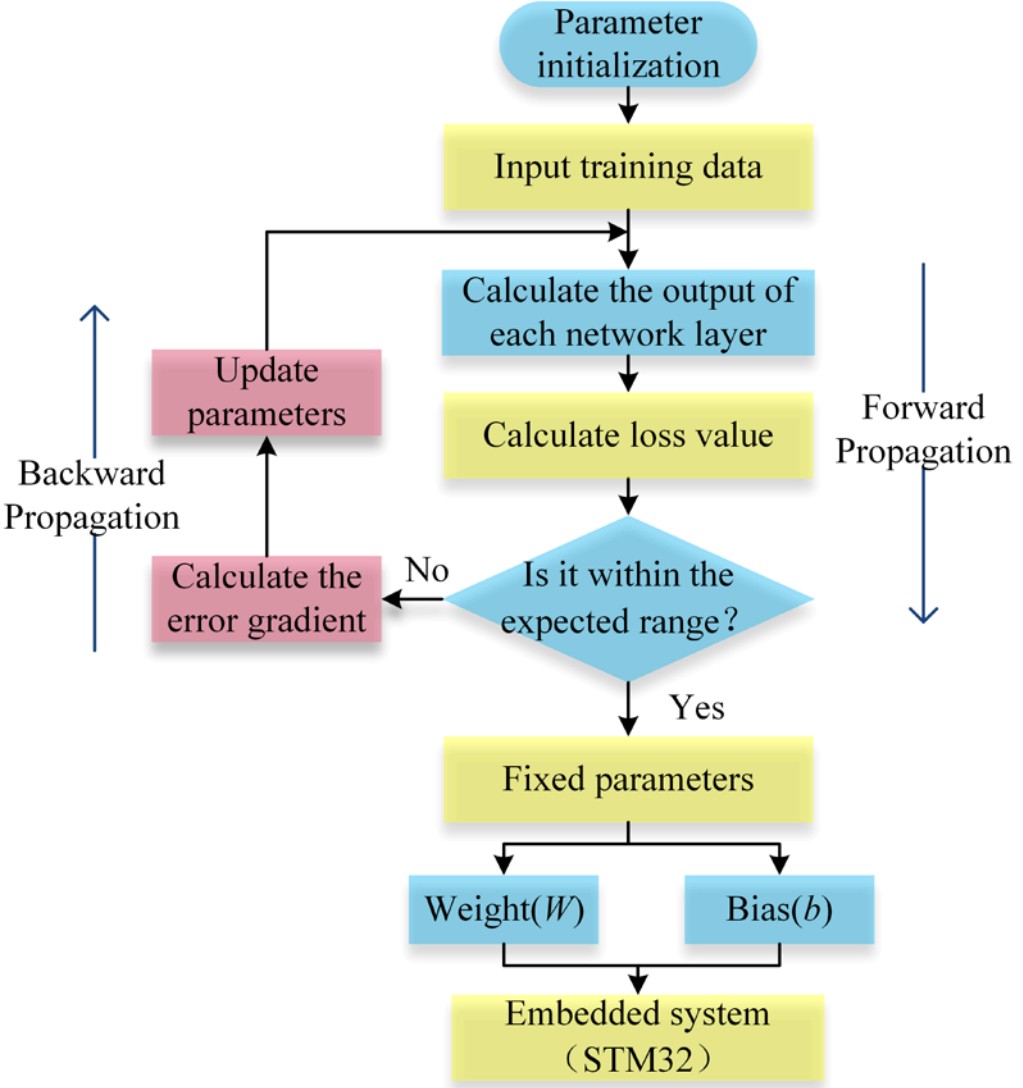

**Figure 3.** Parameter training and transplantation diagram for the proposed model.

*3.1. Parameter Training*

The traditional supervised learning method is utilized for parameter training. In this paper, Adam is selected as the optimizer, and batch gradient descent (BGD) is employed as the optimization method for parameter training. The specific training steps are as follows:

(1) The signal-to-grayscale image conversion method is used to convert the time–domain signals in the adopted dataset into grayscale images with a size of $32 \times 32$, which are randomly divided into training and test datasets with a ratio of 7:3.

(2) All weight matrices, $W$, and biases, $b$, in the network are initialized, and the most commonly used Gaussian distribution is selected for initialization.

(3) The training samples transformed into grayscale images are input into the network, and the output is obtained by forward network propagation.

(4) The loss value of the output layer is calculated based on the loss function.

(5) If the loss value is less than the expected value, the loop is ended, and the best model and parameters are saved; otherwise, backpropagation is conducted.

(6) Backpropagation is used to calculate the loss value of the output layer with the loss function, then the result is returned layer by layer, the error and gradient of each layer are calculated, and the weights and other parameters are updated in the direction of gradient descent, according to the set learning rate. Finally, step (3) is repeated for the next round of the parameter adjustment process.

Based on the above process, it is clear that three key factors must also be determined to be able to effectively train the network: batch size, learning rate, and loss function.

### 3.1.1. Selection of Batch Size

It has been confirmed through engineering practice that optimizing the model convergence speed is best achieved with the use of mini-batch. If the batch is set too small, each input sample will contain too few classes, resulting in a slow model optimization process. On the other hand, if the batch is set too large, it will lead to a local optimum. Therefore, by comparing the training results with different batch sizes multiple times, it was found that a batch size setting of 16 had the best convergence and the highest accuracy rate.

### 3.1.2. Selection of Learning Rate

The learning rate is reflected in the network training process as the step size in the parameter adjustment path. A small learning rate may cause the model to never converge or fall into a suboptimal solution, whereas a large learning rate may speed up the model training in the early stage. However, the value of the loss function may continue to oscillate and wander in the later stage, making it difficult to find the optimal region. Therefore, in order to obtain the best training results, this paper used an automatically decaying learning rate update method with the ReduceLROnPlateau function in Pytorch. This function allows the learning rate to automatically decrease by 50% when the model prediction accuracy does not improve within 10 epochs by setting the parameters.

### 3.1.3. Selection of Loss Function

In machine learning, we need the predicted data distribution learned by the model on the training data to be as similar as possible to the real data distribution. Since its application to the field of machine learning, cross-entropy can well evaluate the difference between the probability distribution obtained from the current training and the real distribution; it can also effectively avoid gradient disappearance. Therefore, cross-entropy was selected as the loss function for this paper, and its function expression is shown in Equation (7) in a $N$ classification task. $N$ is the total number of categories with known bearing status in the classification task.

$$Loss = -\sum_{i=1}^{N} p(x_i) \log q(x_i) \tag{7}$$

where $p(x_i)$ is the target distribution, and $q(x_i)$ is the predicted distribution.

### *3.2. Parameter Transplantation*

The optimal parameters, corresponding to the training dataset, were obtained by PC-based model training. These parameters were saved as text files and imported into the embedded system for reading by the already constructed network model to fill the correct positions of the convolutional kernels and biases in each network layer.

## 4. Experimental Validation

### *4.1. Case Study 1: CWRU Dataset*

In this section, the performance of the proposed model is evaluated using the CWRU dataset and compared with other classification methods.

### 4.1.1. Dataset Description

The dataset for this experiment was provided by Case Western Reserve University (CWRU) [45]. The CWRU fault simulation test platform, as illustrated in Figure 4, was used to collect data from an accelerometer placed at the end of the fan. The bearing type was SKF6203-2RS. Three types of bearing faults were seeded by electrical discharge machining (EDM): inner race faults (IF), ball faults (BF), and outer race faults (OF). Each fault type had three different fault sizes—0.1778 mm, 0.3556 mm, and 0.5334 mm—in addition to providing data for normal conditions, which resulted in a total of 10 types of

motor bearing vibration data. The sampling frequency for the experiment was 12 kHz. The data classification results are presented in Table 2.

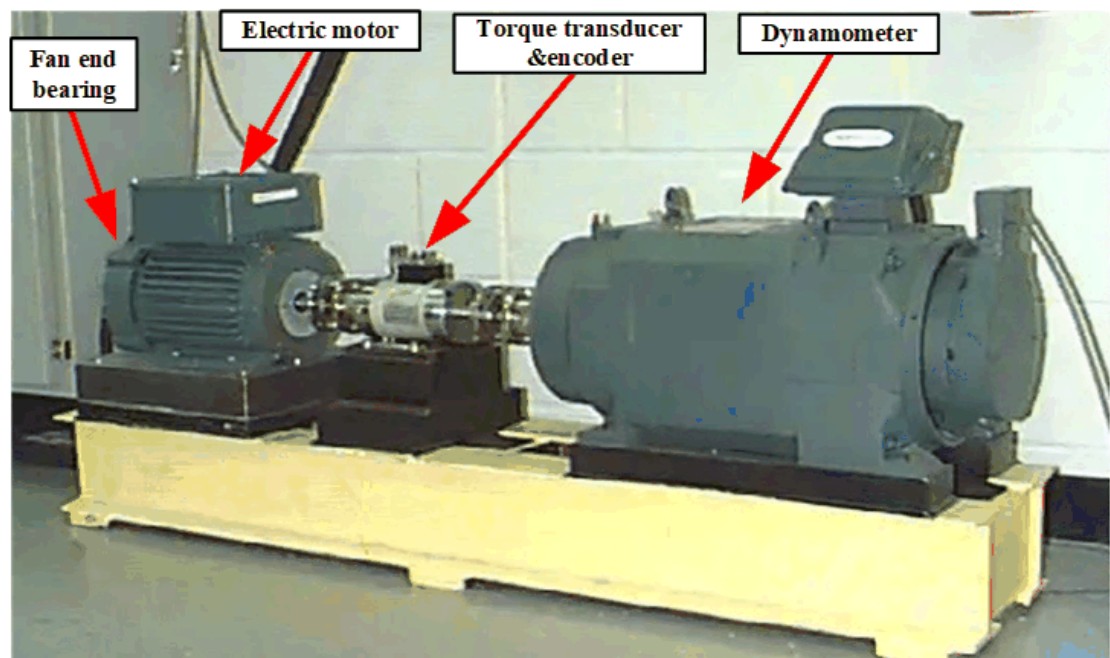

**Figure 4.** The CWRU fault simulation test platform.

**Table 2.** CWRU dataset classification description.

| File Number (.mat) | Fault Location | Fault Size (mm) | Sample Number | Label |
|---|---|---|---|---|
| 98 | Normal | 0 | 100 | 0 |
| 105 | Inner Race | 0.1778 | 100 | 1 |
| 169 | Inner Race | 0.3556 | 100 | 2 |
| 209 | Inner Race | 0.5334 | 100 | 3 |
| 118 | Ball | 0.1778 | 100 | 4 |
| 185 | Ball | 0.3556 | 100 | 5 |
| 222 | Ball | 0.5334 | 100 | 6 |
| 130 | Outer Race | 0.1778 | 100 | 7 |
| 197 | Outer Race | 0.3556 | 100 | 8 |
| 234 | Outer Race | 0.5334 | 100 | 9 |

4.1.2. Image Conversion

The length of the bearing vibration signals mentioned above was 307,200, and each bearing vibration signal type was truncated into 100 sections, each with a length of 1024. Each truncated signal was converted into a grayscale image, according to Equation (1), as depicted in Figure 5. In Figure 5, the corresponding relationships between the different labels and the fault categories of the bearings are shown in Table 2. In total, 1000 grayscale image samples were obtained for the network input, with each image having the size of 32 × 32. Therefore, the dataset contained a total of 10 types, with 100 samples of each type. As Figure 5 shows, although there were significant differences between the grayscale images of the different statuses, it was difficult to identify the type and size of the bearing fault, so the model was still required to make an accurate judgment.

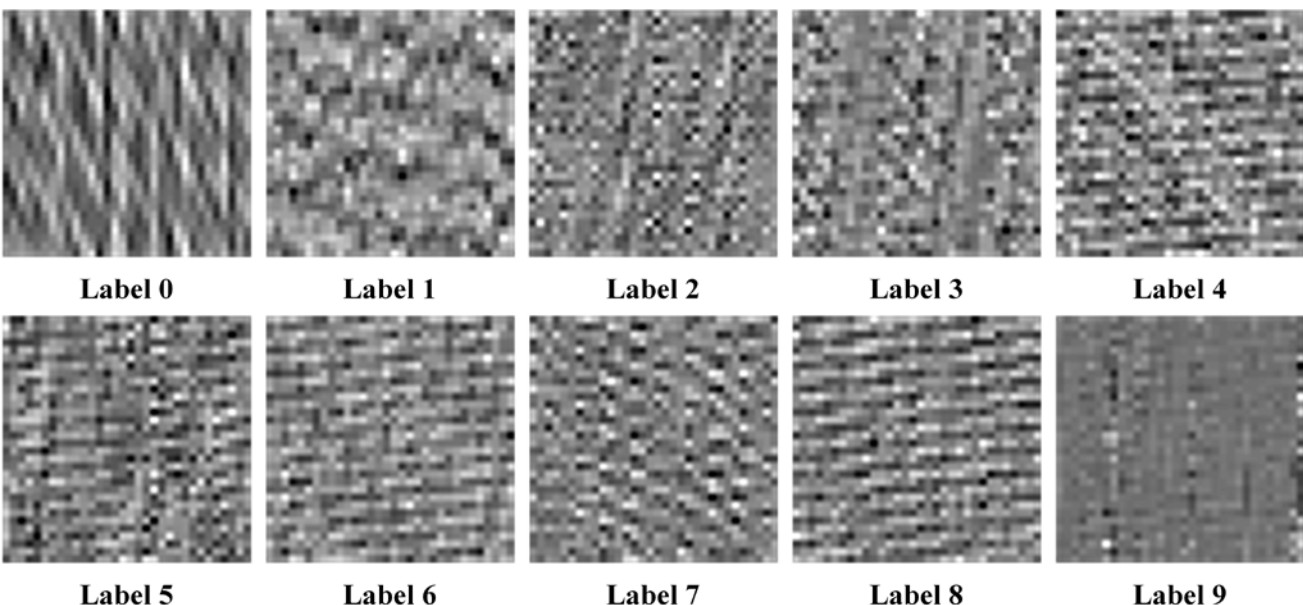

**Figure 5.** Converted grayscale images.

### 4.1.3. Model Training

The experiment was constructed based on the TensorFlow framework. The detailed training environment settings are displayed in Table 3. The ratio of the training set to the test set is 7:3; thus 70% of the 1000 grayscale images were randomly selected for the training set. The mini-batch size was set at 16, and the loss function was the cross-entropy. The learning rate was set to automatic decay during training. If the accuracy did not improve within 10 epochs, the learning rate decreased by 50%. After multiple tests, the training epoch was set to 250, and the max function was set to automatically save the parameters at the highest accuracy during training.

**Table 3.** Experimental environment configuration.

| Experimental Environment | Hardware Configuration |
| --- | --- |
| Operating system | Windows 11 |
| RAM | 16 G |
| CPU | Intel(R)Core(TM)i5-10400 CPU@2.90 GHz |
| GPU | NVIDIA GeForce GTX970 |
| TensorFlow | 2.7.0 |
| Python | 3.8 |

### 4.1.4. CNN Structure Testing Result

To reduce the effect of randomness, a total of 10 experiments were conducted; in each experiment, 30% of the labeled dataset was fed into the model as a test set, which was then validated based on the results of the random division. Table 2 shows the correlation between the predicted labels and the fault size and type. As can be seen in Table 4, the mean accuracy of the ten experiments was 94.40%. Based on the results of one of the experiments, shown in Figure 6, the best outcome was achieved by the model upon reaching 94.33% accuracy for the test set after about 235 iterations. To further visualize the prediction results of the model, the predicted labels and true labels were illustrated using a confusion matrix [46], as seen in Figure 7. As can be seen in the Figure 7, only 17 samples could not be correctly classified in the model test for 300 samples. All the samples labeled with 0,1, and 7 were correctly classified, and a small number of samples labeled with 2, 3, 4, 5, 6, 8, and 9 were incorrectly classified.

**Table 4.** Results of 10 experiments.

| Experiment Number | 1 | 2 | 3 | 4 | 5 | 6 | 7 | 8 | 9 | 10 |
|---|---|---|---|---|---|---|---|---|---|---|
| Accuracy (%) | 94.33 | 93.66 | 95.00 | 93.33 | 96.66 | 95.66 | 94.66 | 91.66 | 95.00 | 94.00 |
| Mean Accuracy (%) | | | | | 94.40 | | | | | |

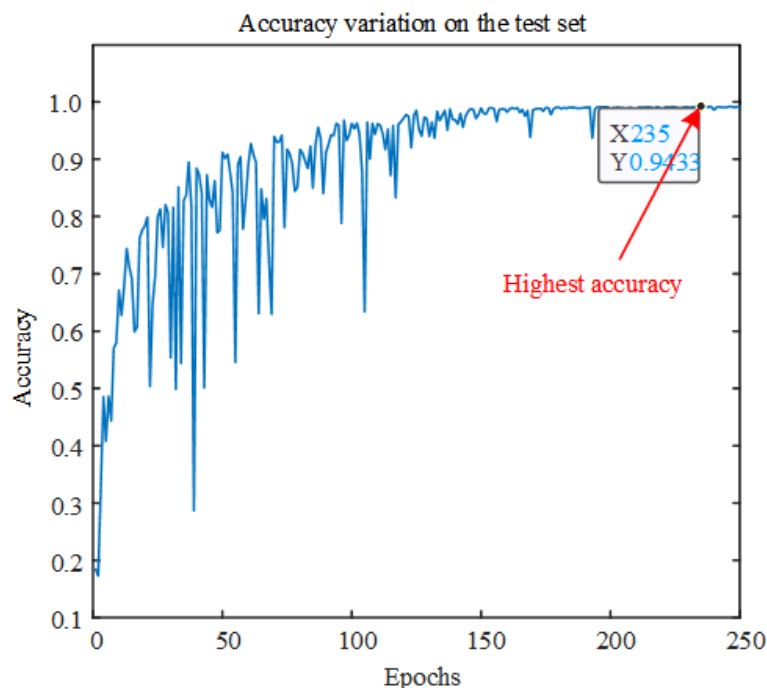

**Figure 6.** Test set accuracy on the CWRU dataset.

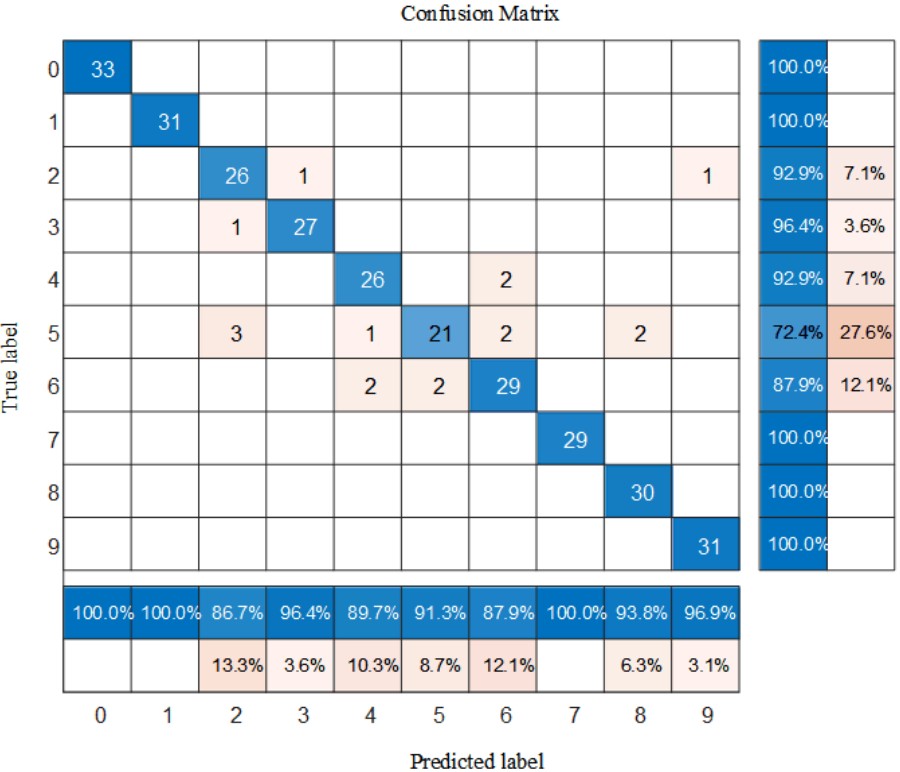

**Figure 7.** Confusion matrix for the CWRU dataset identification results.

### 4.1.5. Comparison with Other Methods

As shown in Table 5, other machine-learning-based fault diagnosis methods were utilized for the evaluation of the performance of the proposed model; the mean prediction accuracy was used as the index for the comparison.

**Table 5.** Performance comparison of the model on the CWRU dataset.

| Methods | Feature Extraction Methods | Training Sets | Testing Sets | Mean Accuracy (%) | Difference Rate (%) |
|---|---|---|---|---|---|
| Machine Learning (ML) | Random Forest | 2000 | 1000 | 75.00 | +25.87 |
| ML | SVM | 2000 | 1000 | 78.90 | +19.65 |
| DL [47] | 1D-CNN | 2000 | 1000 | 99.30 | −4.93 |
| DL [47] | CNN | 2000 | 1000 | 97.80 | −3.48 |
| ADCNN [48] | ADCNN1 + ADCNN2 | 500 | 500 | 99.70 | −5.32 |
| CNN based Markov [49] | CNN + HMM | 9600 | 4800 | 98.13 | −3.80 |
| DBN Based HDN [50] | WPT | 500 | 500 | 99.03 | −4.68 |
| Ensemble CNN and DNN [51] | CNNEPDNN | 2000 | 370 | 97.35 | −3.03 |
| Proposed method | CNN | 700 | 300 | 94.40 | — |

The comparison results in Table 5 show that the proposed method achieved high accuracy just by training a few samples. The last column of Table 5 indicates the difference rate between the proposed method and other methods in terms of prediction accuracy. The calculation equation is as follows:

$$DR = \frac{P_2 - P_1}{P_1} \times 100\% \tag{8}$$

where $P_2$ represents the accuracy of the proposed method, and $P_1$ is the accuracy of the methods used for comparison. The mean prediction accuracy of S-AlexNet is higher than that of the machine-learning-based methods random forest and SVM by 19.4% and 15.5%, respectively. Among the DL-based fault diagnosis methods—1D-CNN, CNN, ADCNN, CNN based Markov, DBN Based HDN, and Ensemble CNN and DNN—the mean accuracy reached 99.3%, 97.8%, 99.7%, 98.13%, 99.03%, and 97.35%, respectively. Although the accuracy rate of S-AlexNet is 94.4%, which is lower than that of the DL-based method mentioned above, the difference rate does not exceed 6%, indicating that the proposed method can accomplish the diagnosis of bearing vibration signals and meet the monitoring requirements of industrial sites.

### 4.2. Case Study 2: Self-Built Dataset

In this section, a dataset, containing seven bearing states, is first obtained using an experimental platform that we built by ourselves, followed by the training of the proposed model using the self-built dataset. Then, the trained model is deployed in the embedded system for the intelligent fault diagnosis of bearings. Finally, the performance of the model on the embedded system is compared with the results of published papers.

### 4.2.1. Experimental Dataset Validation

A. Experimental Testbed

The experimental system used to acquire the bearing data is shown in Figure 8. The system consists of a data acquisition experimental bench, acceleration sensor, NI data acquisition system, and a monitor for displaying the acquisition status. The NSK 6012 bearings were selected for the experiment and were machined to fault using the wire-cutting method, as shown in Figure 9.

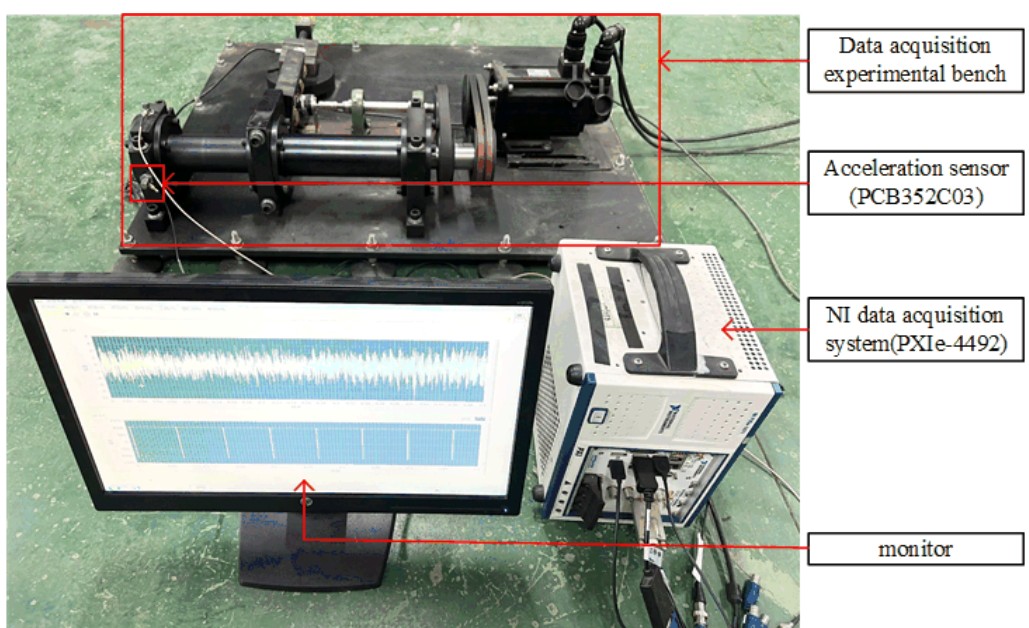

**Figure 8.** Testbed for bearing data acquisition.

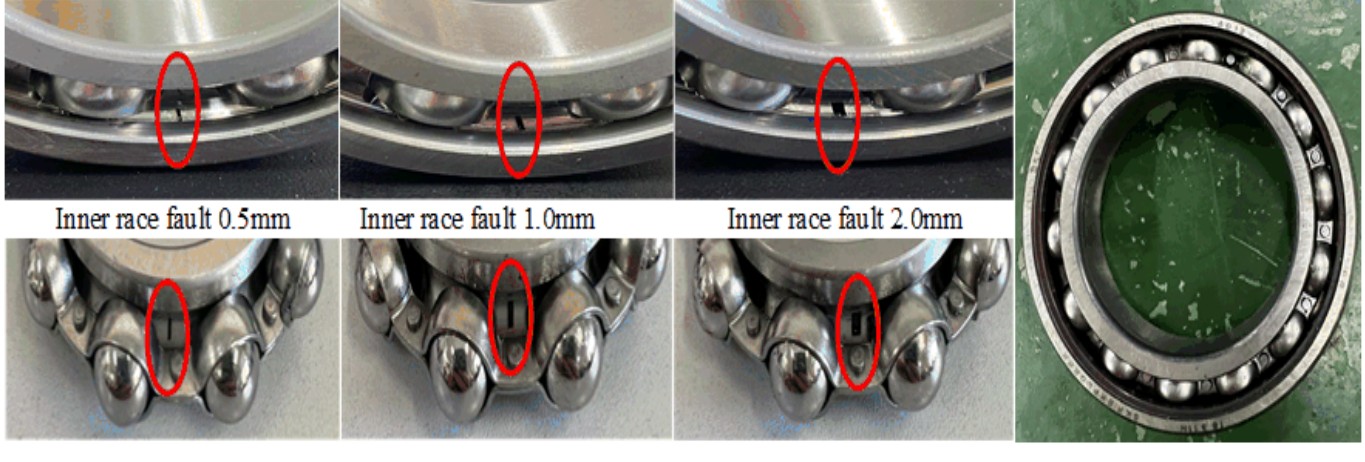

**Figure 9.** Pictures of faulty bearings.

B.   Experimental Work Conditions

The bearing was rotated at 800 rpm. Signal acquisition was conducted using a PCB352C03 single-axis sensor and NI's PXIe-4492 (Austin, TX, USA)acquisition system. The sampling frequency was set to 20 khz. A radial load of 2462 N was applied to the bearing during the operation.

C.   Experimental Dataset Description

As shown in Table 6, the obtained data contained a total of 3500 samples of seven types of normal bearings, inner race failures, outer race failures, and three failure levels. The time domain and envelope spectrum images of the seven data types are sequentially shown in Figure 10. The source code and dataset of Case Study 2 is available at open-source GitHub ([Online]. Available: https://github.com/Solitudeas/ahu_320_dataset.git (accessed on 13 March 2023)).

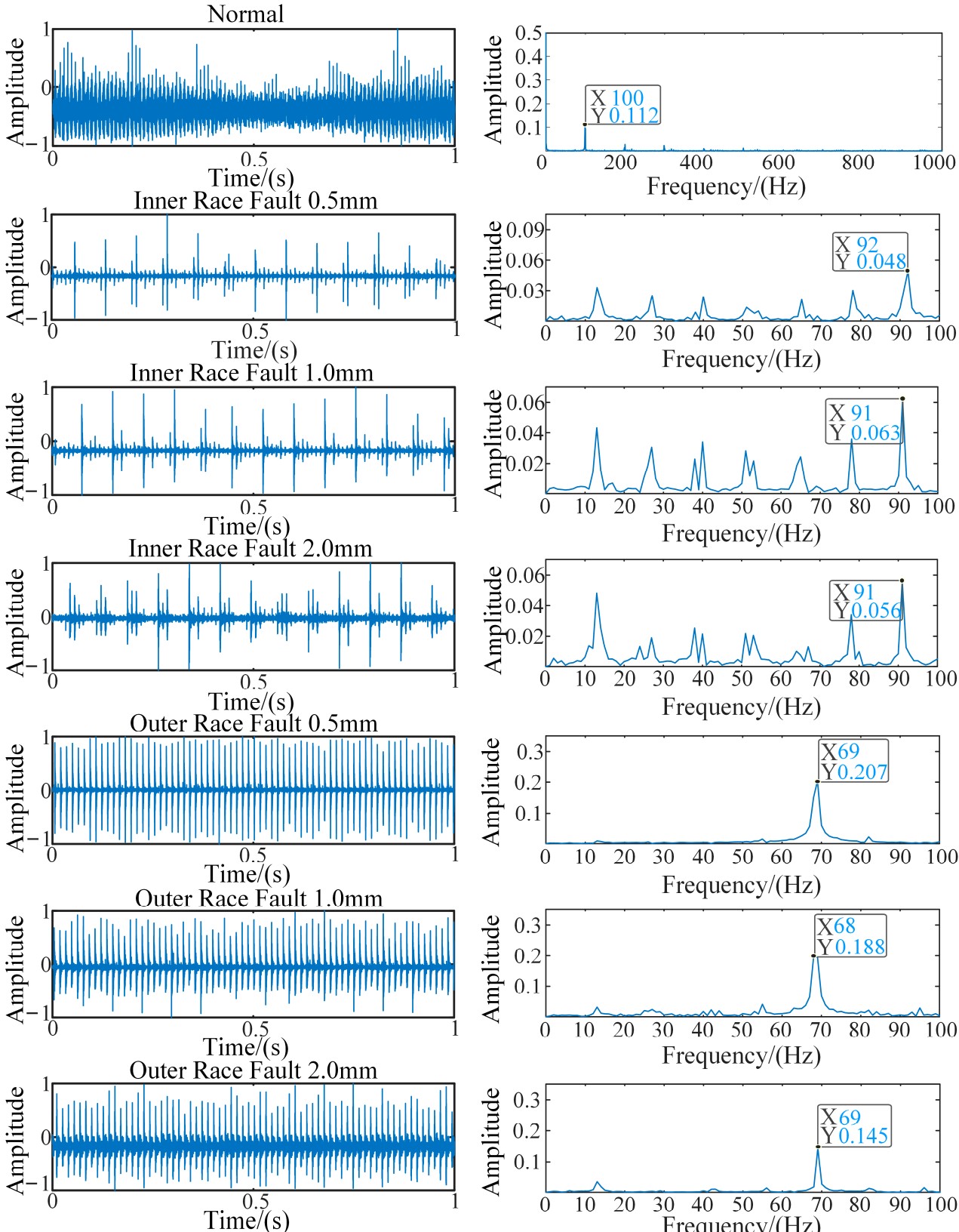

**Figure 10.** Time domain and envelope spectrum images of bearings in seven statuses.

**Table 6.** Self-built dataset classification description.

| Fault Mode | Rotation Speed (rpm) | Radial Load (N) | Sample Number | Label |
|---|---|---|---|---|
| Normal | 800 | 2462 | 500 | 0 |
| Inner Race Fault 0.5 mm | 800 | 2462 | 500 | 1 |
| Inner Race Fault 1.0 mm | 800 | 2462 | 500 | 2 |
| Inner Race Fault 2.0 mm | 800 | 2462 | 500 | 3 |
| Outer Race Fault 0.5 mm | 800 | 2462 | 500 | 4 |
| Outer Race Fault 1.0 mm | 800 | 2462 | 500 | 5 |
| Outer Race Fault 2.0 mm | 800 | 2462 | 500 | 6 |

D.  Image Conversion

Referring to the image conversion process in Case Study 1, the seven experimentally obtained time-domain vibration signals were converted into grayscale images, according to Equation (1). As can be seen in Figure 11, which shows the grayscale images obtained from the conversion of the bearing signals in the seven states, each vibration signal with a length of 1024 was converted into a grayscale image with a size of 32 × 32 as a sample of network training. Ultimately, a total of 3500 grayscale images were obtained by conversion, and the corresponding relationships between the label values of each grayscale image and the bearing fault categories were shown in Table 6.

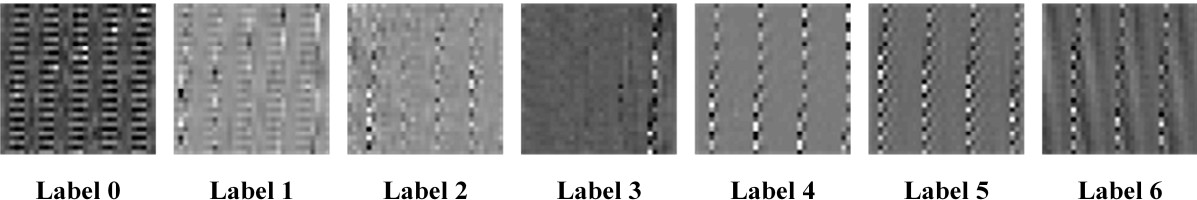

**Label 0**       **Label 1**       **Label 2**       **Label 3**       **Label 4**       **Label 5**       **Label 6**

**Figure 11.** Grayscale images of seven experimental signals.

E.  Model Training and Result

To eliminate the effect of randomness on the results, 10 experiments were conducted using the same training method as that utilized in Case Study 1, and the accuracy curve obtained for one of them is shown in Figure 12. The model reached its optimal state at about 125 iterations, achieving an accuracy of 99.81%. The results of the 10 experiments were shown in Table 7, with an average prediction accuracy of 99.84%.

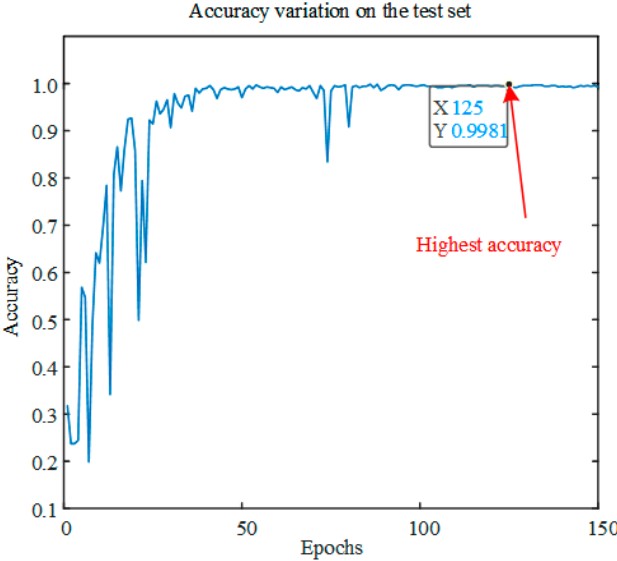

**Figure 12.** Accuracy of the test set on the experimental dataset.

**Table 7.** Results of 10 experiments.

| Experiment Number | 1 | 2 | 3 | 4 | 5 | 6 | 7 | 8 | 9 | 10 |
|---|---|---|---|---|---|---|---|---|---|---|
| Accuracy (%) | 99.81 | 99.90 | 99.71 | 99.90 | 99.81 | 100 | 99.81 | 99.71 | 99.90 | 99.81 |
| Mean Accuracy (%) | | | | | 99.84 | | | | | |

F.    Parameter Transplantation

Through model training, a total of 7399 optimal parameters could be obtained from the PC, as shown in Table 8, and they were transformed into a matrix of the corresponding convolutional kernels and network biases. These matrices were then transplanted to the embedded system in the form of arrays and were read in C to fill the correct positions of the convolutional kernels and biases in each network layer.

**Table 8.** Description of optimal parameters on the self-built dataset.

| Layer | Convolution Kernel Size | Bias Size | Parameters Number (Total of 7399) |
|---|---|---|---|
| Conv1 | $4 \times 10 \times 10$ | 4 | 404 |
| Conv2 | $16 \times 5 \times 5$ | 4 | 404 |
| Conv3 | $32 \times 3 \times 3$ | 8 | 296 |
| Conv4 | $64 \times 3 \times 3$ | 8 | 584 |
| Conv5 | $64 \times 3 \times 3$ | 8 | 584 |
| FC1 | $72 \times 64$ | 64 | 4672 |
| FC2 | $64 \times 7$ | 7 | 455 |

G.    Test Results in the Embedded System

Thirty percent of the labeled dataset taken for classification was imported into the embedded system as a test set for validation. Each sample of the test set was in turn identified by the model in the embedded system to obtain the predicted labels, which were subsequently compared with the true labels and classified as statistically correct or incorrect. The results of the embedded system identification for the 1050 test samples are shown in Figure 13. For further visualization, the predicted labels and the true labels were plotted as confusion matrices, as shown in Figure 14. As can be seen in Figure 14, the transplanted network was tested on 1050 samples, and only two samples were not correctly classified. Based on Equation (9) and Figure 14, the accuracy of this test is 99.81%.

$$\text{Accuracy} = \frac{TS}{TS + FS} \times 100\% \tag{9}$$

where *TS* is the number of correctly identified samples, and *FS* is the number of misidentified samples. The MCU version of the embedded system used for testing was STM32H743 with 2 MB of FLASH and 1060 KB of SRAM. The average time needed to identify a sample was measured to be 583 ms after several tests, which is very short when compared to the time required to upload the data to the server for analysis and return the diagnostic results.

4.2.2. Offline Fault Diagnostic Test

As shown in Figure 15, the platform for conducting off-line bearing fault diagnosis consisted of a shaft system, servo motor, speed encoder, and fault diagnosis acquisition card. The LCD display on the fault diagnosis acquisition card showed the bearing status, the extent of the fault, and the signal's envelope spectrum image at the bottom of the screen in order to present the fault frequency value as a reference. Figure 16 shows the information displayed on the edge-side LCD when the experimental platform was loaded with bearings of different types and degrees of failure. According to Equations (10) and (11), the theoretical inner race fault frequency and outer race fault frequency of the experimental bearing were 91.35 hz and 68.65 hz, respectively. The experiments demonstrated that the model transplanted into the

embedded system could correctly identify the bearing status in the offline state, which verified the feasibility of the proposed method in the field of bearing fault diagnosis.

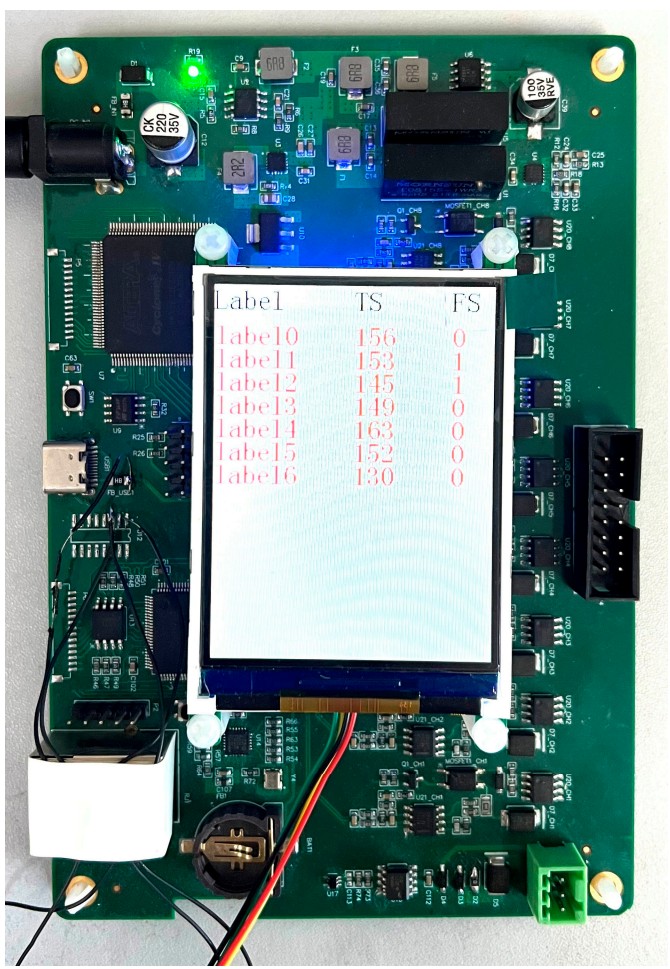

**Figure 13.** Identification results for the self-built dataset in embedded systems.

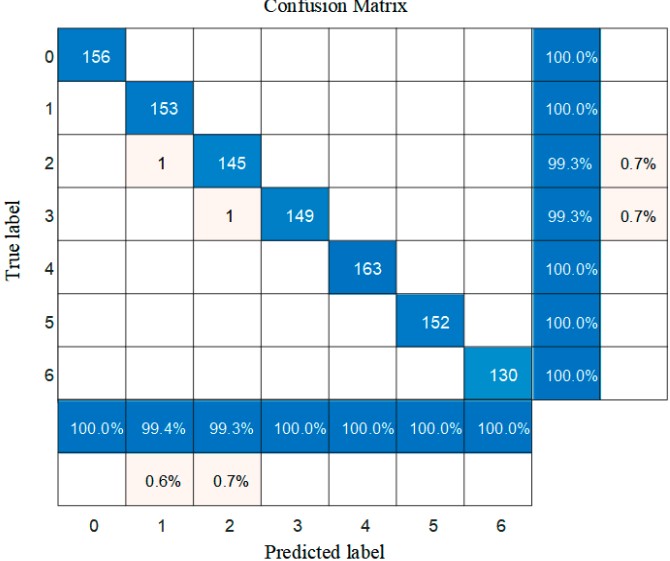

**Figure 14.** Confusion matrix for the result of experimental data identification.

$$f_O = \frac{n}{2}\left(1 - \frac{d}{D_p}\cos\alpha\right)f_r \tag{10}$$

$$f_I = \frac{n}{2}\left(1 + \frac{d}{D_p}\cos\alpha\right)f_r \tag{11}$$

where $n$ represents the number of balls, $d$ represents the diameter of the rolling element, $D_p$ represents the groove section size, $\alpha$ represents the contact angle, $f_r$ is the shaft frequency, $f_O$ is the outer race fault frequency, and $f_I$ is the inner race fault frequency.

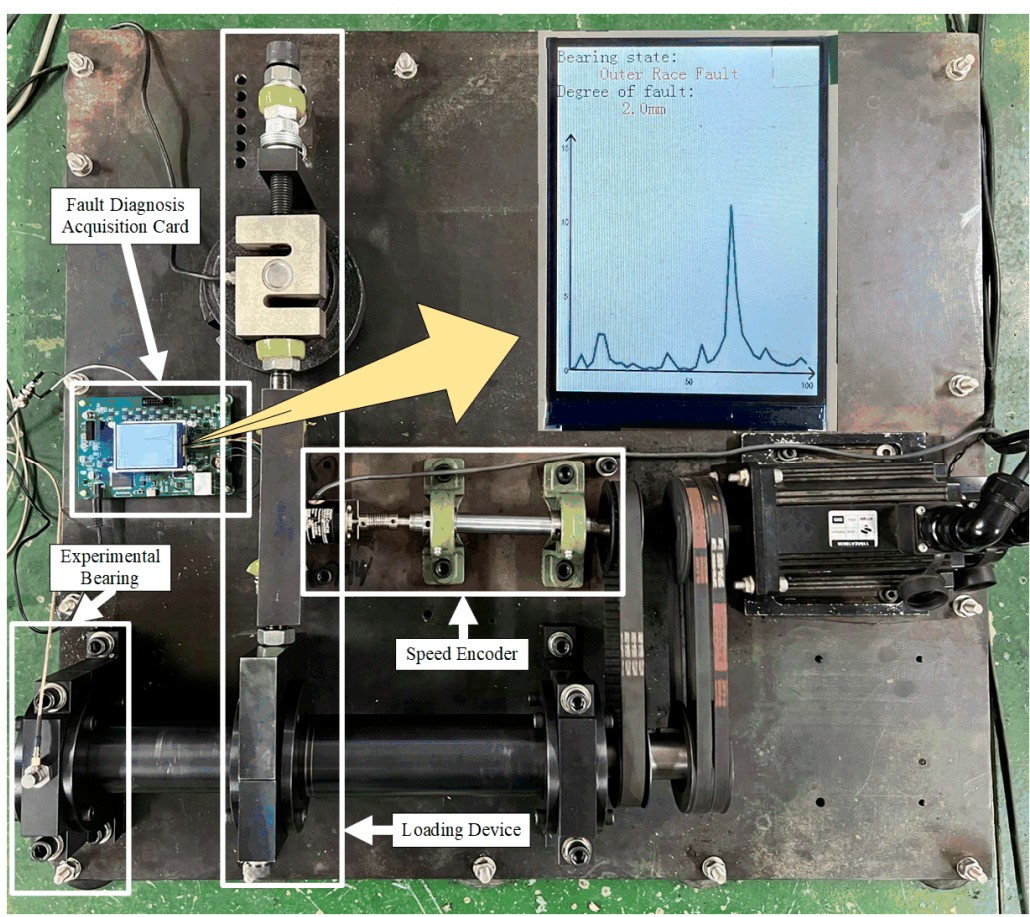

**Figure 15.** Offline fault diagnosis experimental device.

### 4.2.3. Comparative Results of IFD Performance in Embedded Systems

As shown in Table 9, the proposed method in this paper was compared with other edge IFD methods.

Based on the results of comparison with methods 1, 2, and 3, it is clear that the proposed method is beneficial for more practical scenarios, as it did not rely on prior knowledge of fault diagnosis and could directly analyze the collected vibration signals to monitor the bearing status. Compared with 41,304 parameters in method 4 and 110,400 parameters in method 5, the proposed method only had 7399 parameters; the model size was greatly reduced, while the prediction accuracy was also guaranteed, which made the IFD algorithm easier to implement in the MCU of embedded portable devices. FLOPs were used to describe the amount of computation required for a sample to pass through the model, which could reflect the complexity of the model to some extent. Table 9 clearly shows that the complexity of the proposed method is much lower than that of methods 4 and 5, which also used machine learning methods for IFD. The low complexity of the model provided better timeliness for the fault diagnosis of bearings on the edge side and helped

to achieve an earlier detection of abnormal conditions in the equipment. In comparison to other methods, the proposed method had the lowest cost; the model was also constructed using the C language, which is a widely used basic development language in industrial production, especially for embedded devices. Therefore, the method showed an extremely high potential for industrial applications.

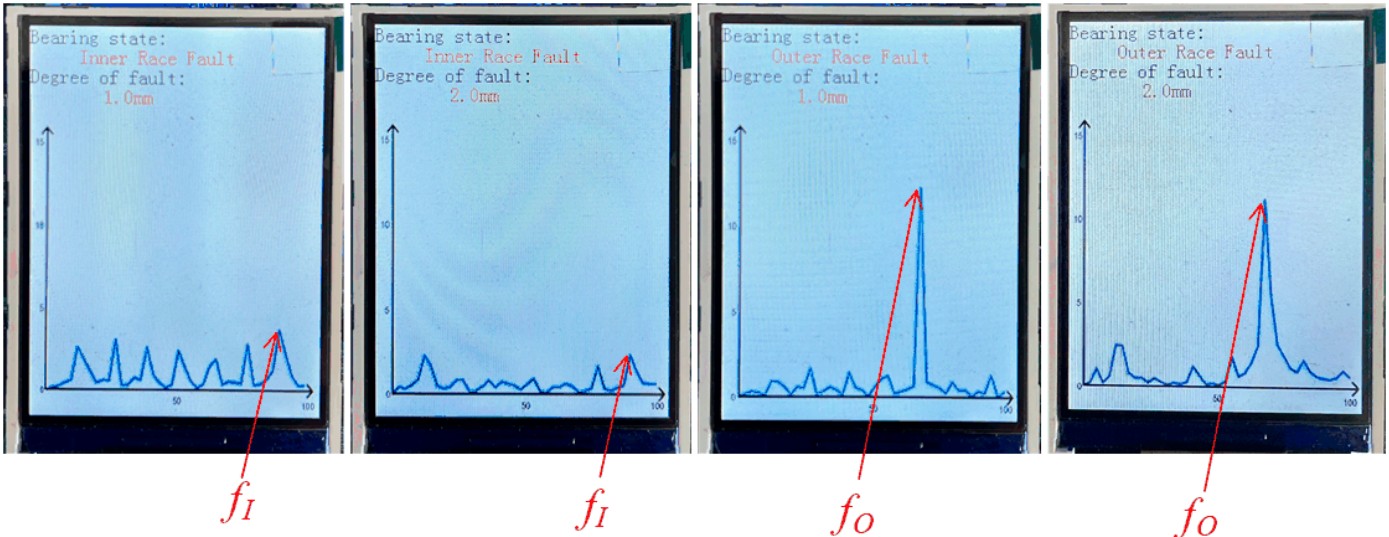

**Figure 16.** Offline bearing diagnostic status display.

**Table 9.** Comparison of different methods.

| Number | Method | Embedded Platform | Prior Knowledge Required | Parameters | FLOPs | Cost (USD) | Language Type |
|---|---|---|---|---|---|---|---|
| 1 | Online order analysis method [31] | STM32F407 + STM32F407 | Yes | — | — | 50.9 | C |
| 2 | Stochastic-Resonance-Based adaptive filter [32] | STM32F407 + STM32F429 | Yes | — | — | 65.1 | C |
| 3 | Hilbert + Random Forest [34] | FPGA: PYNQ-Z2 | Yes | — | 71,680 | 171.2 | Verilog |
| 4 | MobileNet-v2 [33] | Raspberry Pi 3B | No | 41,304 | 1,193,632 | 139.4 | Python |
| 5 | LSTM-based model [35] | Raspberry Pi 3B | No | 110,400 | 776,000 | 139.4 | Python |
| 6 | Proposed method | STM32H743VIT6 | No | 7399 | 589747 | 44.4 | C |

## 5. Conclusions and Future Research

An edge intelligent diagnosis method for bearing faults based on a parameter transplantation CNN was proposed in this paper. A model that fits the small and efficient character of embedded systems was designed and deployed in an embedded system to monitor the bearing status in real time. The method converted the raw vibration signals into grayscale images as model input, which was used directly for the bearing fault diagnosis on the edge side. The model was validated using the CWRU motor bearing dataset with an identification accuracy of 94.40%, and the dataset was collected using our self-built experimental bed with a 99.84% identification accuracy. The comparison of the results with those of published papers demonstrated that the proposed method has the advantages of good real-time performance, high accuracy, portability, and low cost, which compensate for the lack of current PC-based intelligent fault diagnosis methods and provides simple and effective solution for implementing bearing IFD on the edge side. Therefore, the proposed method has the potential for application in industrial production.

The limitations of this method in practical application are the following. Firstly, the types of faults are common ones, and, if a new fault appears, it will be misclassified as a known fault type. Secondly, it is necessary to collect sufficient tagged data each time a new model is acquired, which can be expensive in practical engineering. Further research could be conducted in the following areas based on these limitations. First, the mechanistic

study of a fault should continue in order to find the unknown fault. Second, research on migration learning theory can be conducted to reduce the amount of data that need to be collected in order to train the model and reduce costs.

**Supplementary Materials:** The following supporting information can be downloaded at: https://github.com/Solitudeas/ahu_320_dataset.git (accessed on 13 March 2023).

**Author Contributions:** Conceptualization, X.D. and Y.L.; methodology, X.D. and H.W.; software, Z.H.; validation, X.D. and X.L.; formal analysis, X.D. and Z.H.; investigation, X.D. and H.W.; resources, X.D. and Z.C.; data curation, X.D. and Z.H.; writing—original draft preparation, X.D. and Y.L.; writing—review and editing, X.D.; visualization, X.D. and Z.H.; supervision, Y.L.; project administration, Y.L.; funding acquisition, X.L. and Z.C. All authors have read and agreed to the published version of the manuscript.

**Funding:** This work was supported by the Key Basic Research Project [MKF20210008], the National Natural Science Foundation of China [52075001,52105082,52105040,52275075], and the Natural Science Project of Education Department of Anhui Province [KJ2021A0011].

**Institutional Review Board Statement:** Not applicable.

**Informed Consent Statement:** Not applicable.

**Data Availability Statement:** Data are contained within the article or Supplementary Materials.

**Conflicts of Interest:** The authors declare no conflict of interest.

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
