# Peer review of "An Edge Intelligent Method for Bearing Fault Diagnosis Based on a Parameter Transplantation Convolutional Neural Network"

_electronics, doi:10.3390/electronics12081816_

Round 1

Reviewer 1 Report (Previous Reviewer 1)

The paper notably improved after its revision, and all of my comments were properly and extensively addressed. Good job!

Author Response

    Thank you very much for your approval on our revisions. Your comments are all valuable and very helpful for improving the quality of our manuscript. We would like to thank you again for your time and effort in reviewing our manuscript.

Reviewer 2 Report (Previous Reviewer 3)

Based on the current version of manuscript, I can tell that the authors have put significant efforts to improve the quality of manuscript. Thanks for the efforts made. There are some comments need to be addressed by the authors as follows:

1. Abstract - Please present the qualitative results. 

2. Lines 173 & 174 - The rational of using equation k = sqrt(m) should be further explained. 

3. Some mathematical symbols appear distorted. Please rectify this issue.

4. Section 3.1 - Based on the description of this section, it seems that authors are training the model from scratch. Why not use transfer learning instead since the pretrained network AlexNet is used in this case?

5. Authors claims that this is a lightweight CNN but there is no further justifications of why it is considered as lightweight CNN. What are the essential differences between AlexNet and S-AlexNet? More clarifications are needed to address this issue.

6. It is not mentioned that how many classes of faulty issues to be classified in the proposed methdology. What is the value of N in Line 284?

7. Figures 5 and 11 - Are there any specific meanings of these labels 1, 2 and so on? If yes, please specify the meaning of these lable.s

Author Response

Reviewer 3 Report (New Reviewer)

1. Please shorten the introduction, if possible.

2. Acknowledge the CWRU properly, as acknowledged in:

Sharma, Snehsheel, S. K. Tiwari, and Sukhjeet Singh. "Integrated approach based on flexible analytical wavelet transform and permutation entropy for fault detection in rotary machines." Measurement 169 (2021): 108389.

Round 2

Reviewer 2 Report (Previous Reviewer 3)

Most of the comments are addressed by the authors. Just one more minor revision to be made for previous comment 5:

- Please use a table to visualize between the difference between AlexNet and S-AlexNet in terms of the network architecture layers to provide better clarity to the readers. Please also state the numbers of trainable parameters of both AlexNet and S-AlexNet.

Author Response

This manuscript is a resubmission of an earlier submission. The following is a list of the peer review reports and author responses from that submission.

Round 1

Reviewer 1 Report

See attached file.

Reviewer 2 Report

This work presented a useful study for bearing fault detection. Although there are some merits, the novelty is limited. 

1) The motivation of this work is not clearly. Edge computing is important, but what are the challenges for applying edge computing (i.e., embeded systems) to fault diagnosis?

2) Why the model should be designed like as shown in Fig. 1.

3) How about the comparison with the state-of-the-art methods.

4) To be honest, this is more like a report, not a scientific paper. 

Reviewer 3 Report

1. Abstract is not properly written and did not highlight the novelty and significance of current study The performance of proposed work need to be elaborated more precisely. 

2. The technical writing of this manuscript is concerning. Many typo, poorly constructed sentences and grammatical issues were observed from the current manuscript. Significant improvement in terms of linguistic quality is needed.

3. Problem statements and research gaps that motivated the current study are not evident. The significance of current research work is not clear.

4. The technical contributions and novelty of current research are not evident as well.

5. Literature review related to the fault diagnosis of rolling bearing are completely missing in this manuscript. 

6. One of my main concerns on this manuscript is the novelty and contributions on the proposed methodology. Many important steps including data processing, network training, loss functions and etc. are not described. The parameter training and transplantation mentioned in this manuscript is not adopted from other paper. Hence, I cannot really tell how this manuscript can contribute in enhancing the body of knowledge in this research area.

7. Performance comparison with state-of-art methods are lacking. The performance analyses of proposed algorithm is not convincing. 
